# EvoCodeBench: An Evolving Code Generation Benchmark with Domain-Specific Evaluations

**Jia Li** ♂[1,2]**, Ge Li**[1,2]***, Xuanming Zhang**[3]**, Yunfei Zhao**[1,2]**, Yihong Dong**[1,2]**, Zhi Jin**[1,2]
**Binhua Li**[4]**, Fei Huang**[4]**, Yongbin Li**[4]*
[1]Key Laboratory of High Confidence Software Technologies (Peking University), Ministry of Education
[2]School of Computer Science, Peking University, Beijing, China
[3]Bytedance, [4]Alibaba Group
lijia@stu.pku.edu.cn, lige@pku.edu.cn, shuide.lyb@alibaba-inc.com

## Abstract

How to evaluate Large Language Models (LLMs) in code generation remains an open question. Many benchmarks have been proposed, but they have two limitations, *i.e.,* data leakage and lack of domain-specific evaluation. The former hurts the fairness of benchmarks, and the latter hinders practitioners from selecting superior LLMs for specific programming domains.

To address these two limitations, we propose a new benchmark - **EvoCodeBench**, which has the following advances: ❶ **Evolving data.** EvoCodeBench will be dynamically updated every period (*e.g.,* 6 months) to avoid data leakage. This paper releases the first version - EvoCodeBench-2403, containing 275 samples from 25 repositories. ❷ **A domain taxonomy and domain labels.** Based on the statistics of open-source communities, we design a programming domain taxonomy consisting of 10 popular domains. Based on the taxonomy, we annotate each sample in EvoCodeBench with a domain label. EvoCodeBench provides a broad platform for domain-specific evaluations. ❸ **Domain-specific evaluations.** Besides the Pass@$k$, we compute the Domain-Specific Improvement (DSI) and define LLMs' comfort and strange domains. These evaluations help practitioners select superior LLMs in specific domains and discover the shortcomings of existing LLMs. Besides, EvoCodeBench is collected by a rigorous pipeline and aligns with real-world repositories in multiple aspects (*e.g.,* code distributions). We evaluate 8 popular LLMs (*e.g.,* gpt-4, DeepSeek Coder, StarCoder 2) on EvoCodeBench and summarize some insights. EvoCodeBench reveals the actual abilities of these LLMs in real-world repositories. For example, **the highest Pass@1 of gpt-4 on EvoCodeBench-2403 is only 20.74%.** Besides, we evaluate LLMs in different domains and discover their comfort and strange domains. For example, **gpt-4 performs best in most domains but falls behind others in the Internet domain. StarCoder 2-15B unexpectedly performs well in the Database domain and even outperforms 33B LLMs.** We release EvoCodeBench, all prompts, and LLMs' completions for further community analysis[1].

## 1 Introduction

Large Language Models (LLMs) have shown impressive abilities in code generation [14, 15, 18]. As more and more LLMs emerge, a reliable code generation benchmark is crucial to evaluating and selecting superior LLMs. Many benchmarks have been proposed, such as HumanEval [3], ClassEval

---

*Corresponding authors
[1]https://github.com/seketeam/EvoCodeBench

[7], and DevEval [16]. Researchers spend lots of effort to annotate test data manually and construct these benchmarks. For example, ClassEval and DevEval cost 500 and 674 person-hours, respectively.

Although promising, existing benchmarks have two limitations.

❶ **Data leakage (aka data contamination).** It means that test data is included in the training data. The trained models perform much better on leaked benchmarks than on real-world tasks. Because the training data of LLMs contains almost all open-source code repositories, existing benchmarks probably have data leakages [6]. Researchers have to spend more effort to construct new benchmarks.

❷ **Lack of domain-specific evaluation.** Programming is highly domain-specific. Developers typically focus on specific domains (*e.g.,* database). Compared to comprehensive coding abilities, developers are more concerned about the performance of LLMs in specific domains. However, existing benchmarks lack domain labels or fall into narrow domains. Besides, they ignore domain-specific evaluations and analyses. Thus, the performance of LLMs across domains is still unclear.

**To alleviate the above limitations, we propose a new code generation benchmark named EvoCodeBench.** EvoCodeBench has three novelties. ❶ **Evolving data.** To avoid data leakages, EvoCodeBench is an evolving benchmark and will be dynamically updated every period (*e.g.,* 6 months). Specifically, we build an automatic collection pipeline, which constructs new versions of EvoCodeBench from the latest repositories. More details about the pipeline are in Section 2.3. ❷ **A domain taxonomy and domain labels.** PyPI [24] is a popular open-source community containing code repositories from various domains. Based on the statistics of repositories on PyPI, we design a programming domain taxonomy covering 10 popular domains. Based on the taxonomy, we annotate each sample in EvoCodeBench with a domain label. In the future, we will refine the taxonomy (*e.g.,* adding emerging domains) and provide a broad platform for domain-specific evaluations. ❸ **Domain-specific evaluations.** Besides the Pass@$k$, we propose the Domain-Specific Improvement (DSI), which reflects the position of an LLM in specific domains. Based on the DSI, we define the comfort domains (*e.g.,* DSI > 10%) and strange domains (*e.g.,* DSI < -10%) of LLMs. These metrics allow practitioners to effectively select superior LLMs in specific domains. Model trainers can also discover which domains LLMs are weak to facilitate future iterations.

Besides the above advances, EvoCodeBench has an advantage in data quality. ❹ EvoCodeBench is collected from high-quality open-source repositories. More importantly, EvoCodeBench aligns with real-world repositories in multiple aspects, *e.g.,* code distributions and dependency distributions. This ensures that the performance of LLMs on EvoCodeBench reflects their abilities in real-world development scenarios. ❺ EvoCodeBench offers comprehensive annotations, *e.g.,* natural language requirements, original repositories, reference code, reference dependencies, domain labels, and test cases. EvoCodeBench computes Pass@$k$ and Recall@$k$ to measure the correctness of generated programs in functionality and dependencies.

In this paper, we release the first version - EvoCodeBench-2403, which consists of 275 samples from 25 real-world repositories. Based on EvoCodeBench-2403, we evaluate 8 popular LLMs (*i.e.,* gpt-4 [21], gpt-3.5 [20], DeepSeek Coder [10], StarCoder 2 [19], CodeLLaMa [26]). Based on extensive experiments, we obtain the following insights. ❶ EvoCodeBench significantly alleviates the data leakage and decreases the potential leak rate from 41.47% to 2.18%. ❷ EvoCodeBench provides a reliable evaluation for repo-level code generation. We analyze these LLMs' failed cases and summarize future directions, *e.g.,* long context modeling ❸ We evaluate LLMs in different domains and discover their comfort domains and strange domains. For example, gpt-4 performs best in most domains but performs worse than others in the Internet domain. StarCoder 2-15B unexpectedly performs well in the Database domain and even outperforms 33B LLMs.

## 2 EvoCodeBench

In this section, we first show an overview of EvoCodeBench and then describe its tasks and evaluation metrics. Then, we present the first version - EvoCodeBench-2403 and its statistics. Finally, we describe the automatic pipeline for constructing EvoCodeBench.

### 2.1 Overview

Figure 1 shows a sample in EvoCodeBench. Each sample consists of seven components.

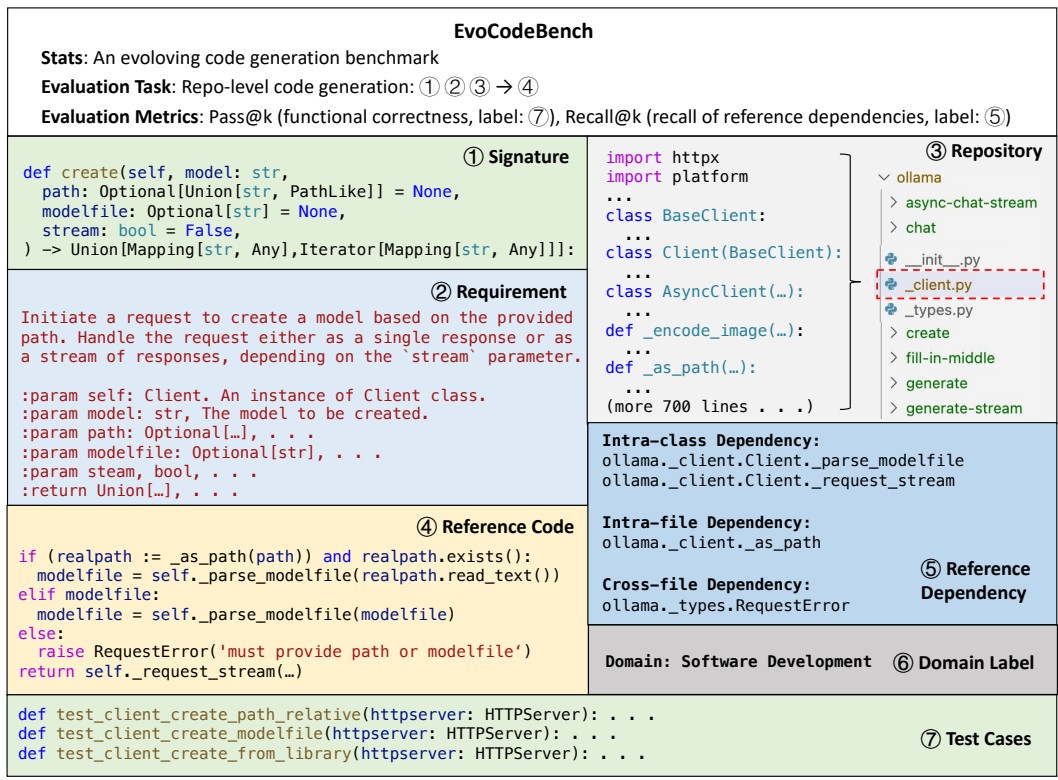

Figure 1: An overview of EvoCodeBench. Each sample consists of seven components.

❶ **Function Signature:** The signature of the target code. ❷ **Requirement:** An English description detailing the functionality of the target code. ❸ **Repository:** The current repository contains hundreds of code files. ❹ **Reference Code:** A developer-written implementation of the target code. This code may invoke dependencies defined in the current repository. ❺ **Reference Dependency:** The dependencies invoked in the reference code include intra-class, intra-file, and cross-file dependencies. ❻ **Domain Label:** The domain of the target code. ❼ **Test Cases:** Test cases are used to check the functional correctness of the code.

## 2.2 Task and Metrics

EvoCodeBench evaluates LLMs in **repo-level code generation**. This task simulates the developers' coding process in a working repository. Given a requirement and a repository, LLMs are tasked to generate the code for the repository. Following previous work [16], EvoCodeBench contains two evaluation metrics, *i.e.,* Pass@$k$ and Recall@$k$.

**Pass@$k$ (Functional Correctness).** Following previous studies [3, 1, 29], we assess the functional correctness of programs by executing test cases and compute the unbiased Pass@$k$. Specifically, we generate $n \geq k$ programs per requirement, count the number of correct programs $c \leq n$ that pass test cases, and calculate the Pass@$k$:

$$\text{Pass@}k := \mathop{\mathbb{E}}_{\text{Requirements}} \left[ 1 - \frac{\binom{n-c}{k}}{\binom{n}{k}} \right] \tag{1}$$

**Recall@$k$ (Recall of Reference Dependency).** We expect that generated programs can invoke relevant dependencies defined in contexts. Following previous work [16], we report Recall@$k$, which computes the recall of reference dependencies in generated programs.

Specifically, LLMs generate $k$ programs per requirement. The dependencies invoked by the $i$-th generated program are denoted as $\mathbb{P}_i$. We compare $\mathbb{P}_i$ with reference dependencies $\mathbb{R}$ and compute the Recall@$k$:

$$\text{Recall@}k := \mathop{\mathbb{E}}_{\text{Requirements}} \left[ \max_{i \in [1,k]} \frac{|\mathbb{R} \cap \mathbb{P}_i|}{|\mathbb{R}|} \right] \tag{2}$$

where $|\cdot|$ means the number of elements of a set.

## 2.3 Benchmark Collection Pipeline

We build an automatic pipeline for collecting EvoCodeBench from the latest repositories. The pipeline consists of four stages as follows.

**Stage I: Repository selection and function scraping.** We crawl high-quality repositories from GitHub, satisfying the following criteria: open-source Python repositories with permissive licenses, created within the last six months, non-fork and non-malicious projects, more than 50 stars, and having explicit unit tests. Then, we extract candidate functions from repositories and exclude trivial functions (*e.g.,* empty or initialization functions).

**Stage II: Execution-based filtering.** For each candidate function, we extract test cases invoking it from current repositories. We use `pip` [22] to install execution environments and leverage `Pytest` [25] to run test cases. Candidate functions without executable test cases are excluded.

**Stage III: Automatic annotations.** We leverage a static analysis-based parser [23] to extract each candidate function's signature, function body (*i.e.,* reference code), and invoked dependencies (*i.e.,* reference dependencies). Because manually writing requirements is laborious, we use LLMs to generate requirements. Specifically, we craft a one-shot prompt, which teaches LLMs to write requirements in a specific format (*i.e.,* functional descriptions and input-output parameters). The prompt template is in Appendix E.2.

Next, we annotate each sample's domain label. To standardize the domains, we manually design a domain taxonomy. Specifically, we collect the statistics (*e.g.,* stars and domains) of repositories in a popular software community - PyPI [24]. Based on the statistics, we determine the top 10 domains with the most high-star repositories and construct the taxonomy. The 10 domains cover most of the repositories on PyPI and are shown in Table 1. In the future, we will continuously refine the taxonomy (*e.g.,* adding emerging domains). Finally, we make a prompt and leverage LLMs to automatically annotate domain labels based on candidate functions and our taxonomy. Functions that do not satisfy any of the domains in our taxonomy are excluded. The prompt template is in Appendix E.2.

In Section 4, we conduct a human evaluation to assess auto-generated annotations. The results show that auto-generated annotations are comparable to human-written ones in most cases (*i.e.,* requirement: 96.7% samples and domain label: 98.5% samples).

Table 1: The domain distribution of EvoCodeBench-2403.

| Domain | Count |
|---|---|
| Scientific Engineering | 120 |
| Software Development | 50 |
| Multimedia | 32 |
| Database | 18 |
| System | 17 |
| Internet | 15 |
| Text Processing | 12 |
| Communications | 8 |
| Utilities | 2 |
| Security | 1 |

**Stage IV: Benchmark Construction.** Finally, we randomly select several candidate functions to construct EvoCodeBench. Following the related work [16], we strive to make EvoCodeBench satisfy the following goals: consistent with the code distribution observed in 500 real-world repositories, close to the average number of dependencies in 500 real-world repositories, including as many samples as possible. We have anonymized all personal information in the benchmark.

## 2.4 EvoCodeBench-2403

Through the above pipeline, we collect and release the first version - EvoCodeBench-2403. The statistics of EvoCodeBench-2403 are shown in Table 2. We discuss its features as follows.

❶ **Latest repositories to avoid data leakage.** Considering that most code LLM's [19, 10] training data is up to September 2023, existing benchmarks might have been leaked. For example, all repositories in CoderEval were created before September 2023. In contrast, the 25 repositories in EvoCodeBench-2403 were created between October 2023 and March 2024 and are not included in the training data. The details of 25 repositories is in Appendix D.2.

Table 2: The comparison between existing benchmarks and EvoCodeBench-2403. `SA` and `Depend` are the abbreviations of "standalone" and "dependency", respectively.

| Benchmark | | Code Distribution | | | Dependency Distribution | | Annotation |
|---|---|---|---|---|---|---|---|
| | #Repo. | #Sample | SA | Non-SA | #Type | #Avg. | |
| CoNaLa [28] | – | 500 | 100% | 0% | 0 | 0 | NL, Code |
| HumanEval [3] | – | 164 | 100% | 0% | 0 | 0 | NL, Code |
| MBPP [1] | – | 974 | 100% | 0% | 0 | 0 | NL, Code |
| APPS [11] | – | 5,000 | 100% | 0% | 0 | 0 | NL, Code |
| PandasEval [30] | – | 101 | 100% | 0% | 0 | 0 | NL, Code |
| NumpyEval [30] | – | 101 | 100% | 0% | 0 | 0 | NL, Code |
| AixBench [17] | – | 175 | 100% | 0% | 0 | 0 | NL, Code |
| ClassEval [7] | – | 100 | 100% | 0% | 0 | 0 | NL, Code, Depend. Name |
| Concode [12] | – | 2,000 | 20% | 80% | 1 | 1.23 | NL, Code |
| CoderEval [29] | 43 | 230 | 36% | 64% | 3 | 1.73 | NL, Code, Depend. Name |
| DevEval [16] | 117 | 1,874 | 27% | 73% | 3 | 3.41 | NL, Code, Depend, Repo |
| EvoCodeBench-2403 | 25 | 275 | 27% | 73% | 3 | 3.46 | NL, Code, Depend Repo, Domain |
| 500 Real Repositories | 500 | 1M+ | 27% | 73% | 3 | 3.22 | – |

❷ **Diverse domains.** EvoCodeBench-2403 covers all programming domains in our taxonomy. The domain distribution of EvoCodeBench-2403 is shown in Table 1. It provides a broad platform to evaluate and analyze the performance of LLMs across domains. Because that EvoCodeBench-2403 is our first version, the domain distribution may be unbalanced. In the future, we will collect new samples from the latest repositories and expand the data sizes in different domains.

❸ **High data quality.** EvoCodeBench-2403 is collected by a rigorous pipeline and contains high-quality test data. First, as shown in Table 2, EvoCodeBench-2403 aligns with real-world repositories in multiple aspects. For example, the code distribution of EvoCodeBench-2403 is consistent with that of 500 real-world repositories[2]. Second, EvoCodeBench-2403 provides comprehensive annotations (*e.g.,* requirements, reference code, reference dependency, and the original repository). These annotations offer a broad arena to explore repo-level code generation. Third, each sample in EvoCodeBench-2403 is equipped with an average of 6 test cases rigorously validated through human reviews. In comparison, each sample in a popular benchmark - MBPP [1] has three test cases on average.

## 3 Experiments

### 3.1 Studied LLMs

We select 8 popular LLMs and evaluate them in EvoCodeBench. They cover general LLMs (*i.e.,* gpt-4-turbo-1106 [21] and gpt-3.5-turbo-1106 [20]) and Code LLMs (*i.e.,* StarCoder 2-{15B, 7B} [19], DeepSeek Coder-{33B, 6.7B} [10], and CodeLLaMa-{13B, 7B} [26]). We use official interfaces or implementations to reproduce these LLMs. We run these LLMs on 4 NVIDIA A100-40GB GPUs.

### 3.2 Data Leakage Detetion

As stated in Section 2, an advantage of EvoCodeBench is to alleviate data leakage significantly. To validate this point, we use the latest data leakage detection approach - CDD [6] to check EvoCodeBench-2403. CDD can detect whether LLMs have been trained on specific benchmarks and their variants. The detection results are shown in

Table 3: The results of data leakage detection.

| Benchmark | LLMs | Leak Ratio (%) ↓ |
|---|---|---|
| HumanEval | gpt-3.5 | **41.47** |
| EvoCodeBench-2403 | gpt-4 | 2.18 |
| | gpt-3.5 | 1.75 |
| | DeepSeek Coder-33B | 1.88 |
| | DeepSeek Coder-7B | 1.82 |
| | StarCoder 2-15B | 1.45 |
| | StarCoder 2-7B | 1.09 |
| | CodeLLaMa-13B | 0.82 |
| | CodeLLaMa-7B | 0.73 |

Table 3. Compared to a mainstream benchmark - HumanEval [3], the leakage rate of EvoCodeBench-2403 drops significantly to less than 3%. Besides, since different repositories typically contain similar programs (*e.g.,* logging functions), it is almost impossible to achieve a 0% leakage rate.

Thus, we think that EvoCodeBench-2403 is leakage-free and can provide trustworthy evaluations in repo-level code generation.

---

[2]We reuse the statistics of 500 real-world repositories reported in related work [16].

Table 4: Pass@$k$ and Recall@$k$ of LLMs on EvoCodeBench-2403. Bold and underlined data indicate top-1 and top-2 results, respectively.

| LLMs | Size | Pass@1 | Pass@3 | Pass@5 | Pass@10 | Recall@1 | Recall@3 | Recall@5 | Recall@10 |
|---|---|---|---|---|---|---|---|---|---|
| Local File (Infilling) | | | | | | | | | |
| gpt-4 | N/A | **20.73** | **23.03** | **24.11** | **25.34** | 68.24 | 70.63 | 72.05 | 73.52 |
| gpt-3.5 | N/A | 17.82 | 21.78 | 23.06 | 24.46 | 61.94 | 68.13 | 69.69 | 70.85 |
| DeepSeek Coder | 33B | 19.64 | 22.78 | 24.29 | 26.01 | **71.46** | **79.93** | **82.11** | **86.25** |
| DeepSeek Coder | 6.7B | 17.82 | 21.02 | 22.40 | 23.97 | 69.58 | 74.04 | 78.00 | 83.22 |
| StarCoder 2 | 15B | 15.27 | 17.54 | 18.63 | 20.09 | 50.90 | 53.29 | 55.89 | 61.76 |
| StarCoder 2 | 7B | 14.91 | 17.29 | 18.63 | 19.86 | 56.35 | 60.59 | 63.74 | 74.20 |
| Local File (Completion) | | | | | | | | | |
| gpt-4 | N/A | **17.45** | **19.65** | **20.80** | **22.41** | 63.49 | 68.67 | 70.00 | 72.07 |
| gpt-3.5 | N/A | 15.64 | 17.29 | 18.21 | 19.36 | 61.44 | 66.25 | 66.82 | 69.89 |
| DeepSeek Coder | 33B | 14.18 | 17.57 | 18.66 | 19.95 | 66.90 | **72.83** | 74.40 | 80.02 |
| DeepSeek Coder | 6.7B | 13.45 | 17.10 | 18.81 | 21.07 | 65.76 | 72.32 | 75.61 | 78.45 |
| StarCoder 2 | 15B | 13.82 | 15.44 | 17.84 | 19.59 | **68.55** | 71.37 | 74.76 | 77.70 |
| StarCoder 2 | 7B | 13.45 | 15.15 | 16.18 | 17.65 | 62.93 | 69.85 | 73.54 | 78.40 |
| CodeLLaMa | 13B | 12.73 | 15.78 | 16.86 | 18.19 | 63.34 | 71.26 | **76.43** | 80.11 |
| CodeLLaMa | 7B | 12.73 | 15.33 | 16.00 | 16.93 | 63.33 | 69.79 | 71.91 | 76.50 |
| Without Context | | | | | | | | | |
| gpt-4 | N/A | **7.27** | **10.05** | **10.70** | 11.49 | 21.58 | 23.93 | 25.69 | 26.23 |
| gpt-3.5 | N/A | 6.55 | 7.85 | 8.28 | 8.73 | 21.66 | 24.31 | 24.77 | 25.40 |
| DeepSeek Coder | 33B | 6.91 | 8.92 | 9.79 | 11.03 | **27.67** | **32.73** | 34.92 | 37.76 |
| DeepSeek Coder | 6.7B | 5.82 | 8.56 | 9.67 | 11.26 | 25.89 | 32.06 | **35.59** | **38.33** |
| StarCoder 2 | 15B | 6.18 | 8.77 | 9.95 | **11.53** | 24.03 | 29.86 | 33.62 | 36.91 |
| StarCoder 2 | 7B | 5.82 | 6.72 | 7.43 | 8.62 | 27.39 | 32.60 | 34.88 | 36.81 |
| CodeLLaMa | 13B | 5.45 | 7.38 | 8.37 | 9.95 | 25.52 | 31.28 | 33.66 | 36.36 |
| CodeLLaMa | 7B | 5.45 | 6.94 | 7.75 | 9.03 | 26.97 | 31.17 | 34.08 | 36.82 |

## 3.3 Performance in Repo-level Code Generation

**Experimental Settings.** Repo-level code generation takes a requirement and a repository as inputs. Typically, a repository is very long and surpasses the context windows of existing LLMs. Following previous work [16, 5], we extract parts of code contexts from the repository as inputs and design the following experimental settings. ❶ **Without context.** We ignore contexts and directly generate the code based on requirements and signatures. ❷ **Local File (Completion).** The local file denotes the code file where the reference code is in. This setting simulates the scenario where developers continue to write code at the end of a file. Besides the requirements and signatures, LLMs can access code contexts above the reference code in the local file. ❸ **Local File (Infilling).** This setting simulates the scenario where developers infill code in the middle of a file. Besides requirements and signatures, LLMs can see the code contexts above and below the reference code in the local file.

We use Pass@$k$ and Recall@$k$ (see Section 2.2) to assess generated programs. In this paper, $k \in [1, 3, 5, 10]$. When $k = 1$, we use the greedy search and generate a single program per requirement. When $k > 1$, we use the nucleus sampling with a temperature 0.4 and sample 20 programs per requirement. We set the top-$p$ to 0.95 and the max generation length to 500. Because EvoCodeBench is an evolving benchmark, this paper evaluates LLMs upon EvoCodeBench-2403. Note that the Pass@$k$ and Recall@$k$ between different versions of EvoCodeBench are not comparable.

**Results.** The Pass@$k$ and Recall@$k$ of studied LLMs are shown in Table 4.

**Compared to previous benchmarks, these LLMs' performance in EvoCodeBench-2403 drops dramatically.** For example, the highest Pass@1 scores of gpt-4 on the latest repo-level benchmark [16] is 53.04. In contrast, gpt-4 only achieves 20.73 on Pass@1 upon EvoCodeBench-2403. The decreases demonstrate that EvoCodeBench is more challenging, and previous benchmarks might have been leaked.

**LLMs benefit from more code contexts in repo-level code generation.** As shown in Table 4, after introducing the contexts, the Pass@$k$ and Recall@$k$ of LLMs obviously increase. For example, the Pass@1 of gpt-4 is improved by 104% and 152% in two settings, respectively. Similar phenomena occur in previous studies [16]. We attribute the improvements to the domain knowledge contained in contexts. Figure 2 shows a uniquely successful case in the Local File (Completion) setting. Without context, gpt-4 fabricated a non-existent field as cache directories, generating the incorrect code. In

*Code Contexts (above):*

```
import concurrent
import json
def _get_num_nodes() -> int:
                    . . .
def _get_cache_dir(name: Optional[str] = None):
                    . . .
def _get_cache_data_dir(name: Optional[str] = None):
                    . . .
```

*Input Code:*

```
def _cleanup_cache(self) -> None:
    """ Clean up cache directories by removing them if
    they exist to prevent issues from corrupted files
    from previous runs, and then recreates these
    directories to ensure they are available for use.
    :param self: DataProcessor. ...
    :return: No return values.
    """
```

*gpt-4-turbo-1106 (without context):*

```
for cache_dir in self.cache_directories: ✘
# Check if the cache directory exists
    if os.path.exists(cache_dir):
      . . .
```
Non-existent attribute

*gpt-4-turbo-1106 (local completion):*

```
# Define the cache directories
cache_data_dir = _get_cache_data_dir()  ✔
if os.path.exists(cache_data_dir):
    . . .
```
APIs defined in contexts

Figure 2: A uniquely successful case in Local File (Completion) setting.

Table 6: The Pass@1 of studied LLMs in different domains upon EvoCodeBench-2403.

| Domain | gpt-4 | gpt-3.5 | DeepSeek Coder 33B | 6.7B | StarCoder 2 15B | 7B | CodeLLaMa 13B | 7B |
|---|---|---|---|---|---|---|---|---|
| Database | **38.89** | **38.89** | 33.33 | 33.33 | **38.89** | 33.33 | 33.33 | 33.33 |
| System | **35.29** | **35.29** | **35.29** | **35.29** | 29.41 | 29.41 | 23.53 | **35.29** |
| Software Development | **12.00** | **12.00** | 8.00 | **12.00** | 10.00 | 6.00 | 8.00 | 8.00 |
| Internet | 20.00 | **26.67** | **26.67** | **26.67** | 20.00 | **26.67** | **26.67** | **26.67** |
| Scientific Engineering | **11.67** | 10.00 | 10.00 | 6.67 | 8.33 | 9.17 | 7.50 | 8.33 |
| Multimedia | **25.00** | 18.75 | 15.63 | 15.63 | 18.75 | 18.75 | 18.75 | 12.50 |
| Text Processing | **16.67** | 0.00 | 0.00 | 0.00 | 0.00 | 0.00 | 0.00 | 0.00 |
| All Domains | 17.45 | 15.64 | 14.18 | 13.45 | 13.82 | 13.45 | 12.73 | 12.73 |

fact, two functions for returning the cache directories are available in the local file. After introducing the local file, gpt-4 successfully invokes relevant functions and generates the correct code.

**Retrieval-Augmented Generation (RAG).** RAG enhances generative models with retrieved information and has achieved promising results in code generation [17, 18]. We apply RAG to repo-level code generation and consider the current repository a retrieval corpus. Because most programs in repositories are not equipped with documentation, we retrieve top-$k$ (*i.e.,* $k = 5$ in this paper) functions with similar names to the target function. Specifically, we split names into tokens based on underscore or camelcase formatting and then match the tokens of names. Finally, we use similar functions as contexts in prompts. The results are shown in Table 5. The performance of both LLMs is improved after introducing similar functions. We attribute the improvements to relevant algorithms and dependencies in similar functions. It inspired researchers to explore more advanced RAG techniques to improve repo-level code generation.

Table 5: Performance of RAG.

| LLMs | Setting | Pass@1 | Recall@1 |
|---|---|---|---|
| gpt-4 | Without Context | 8.31 | 21.08 |
| | Similar Functions | 12.29 | 45.14 |
| gpt-3.5 | Without Context | 6.64 | 21.16 |
| | Similar Functions | 11.62 | 41.93 |

**Error Analyses.** The Pass@$k$ of existing LLMs in repo-level code generation is still low. To determine LLMs' shortcomings, we manually analyze 50 error cases of gpt-4 in the Local File (Infilling) setting. We found the most cases (29 cases) are caused by implementation logic errors. 20 cases failed since the necessary contexts were missing, *e.g.,* APIs defined in other files. Besides, one case failed because of the vague requirement. It shows that existing LLMs' reasoning and coding abilities need to be improved. Meanwhile, how to utilize more contexts is necessary to explore.

## 3.4 Performance in Different Domains

We divide EvoCodeBench into multiple subsets according to the domain labels and then calculate the Pass@1 of LLMs in different domains. The results are shown in Table 6. We ignore three domains with less than 10 samples and leave them for future work.

**EvoCodeBench shows superior LLMs in specific domains.** The Pass@1 scores in overall benchmarks demonstrate the comprehensive coding abilities of LLMs. Because developers typically focus

Table 7: The Domain-Specific Improvements (%) of LLMs in different domains. The comfort domains and strange domains are marked in bleu and red, respectively.

| Domain | gpt-4 | gpt-3.5 | DeepSeek Coder 33B | DeepSeek Coder 6.7B | StarCoder 2 15B | StarCoder 2 7B | CodeLLaMa 13B | CodeLLaMa 7B |
|---|---|---|---|---|---|---|---|---|
| Database | 10.21 | 10.21 | -7.14 | -7.14 | -7.14 | 10.21 | -7.15 | -7.15 |
| System | 9.51 | 9.52 | 9.51 | 9.51 | -11.42 | -11.42 | -42.84 | 9.52 |
| Software Development | 23.81 | 23.81 | -21.43 | 23.81 | -66.67 | 5.71 | -21.43 | -21.43 |
| Internet | -28.59 | 7.15 | 7.15 | 7.15 | 7.15 | -28.59 | 7.15 | 7.15 |
| Scientific Engineering | 26.55 | 11.90 | 11.90 | -39.22 | 2.63 | -8.63 | -22.23 | -8.63 |
| Multimedia | 32.14 | 4.75 | -17.11 | -17.11 | 4.75 | 4.75 | 4.75 | -50.01 |
| Text Processing | 100.00 | -100 | -100 | -100 | -100 | -100 | -100 | -100 |

on specific programming domains, they are more concerned about the performance of LLMs in specific domains. Imagine we are developers focused on internet-related programming tasks. Based on the overall Pass@1, we would think StarCoder 2-7B is stronger than DeepSeek Coder-6.7B, *i.e.,* 13.82 > 13.45. However, according to Table 6, DeepSeek Coder-6.7B performs better than StarCoder 2-7B in the Internet domain. This result can help us to select more suitable models.

**EvoCodeBench uncovers the comfort domains and strange domains of specific LLMs.** For ease of observation, we compute the Domain-Specific Improvement (DSI) of LLMs in different domains. The DSI refers to the average relative improvement of Pass@1 of an LLM in a domain compared to other LLMs. Suppose we evaluate $N$ LLMs on a specific domain, and their Pass@1 scores are represented as $\mathbf{P}$. Then, the DSI (%) of $i$-th LLM in this domain is computed as:

$$DSI_i = \frac{1}{N-1} \sum_j \frac{P_i - P_j}{P_i} * 100 \quad (i \neq j) \tag{3}$$

The larger the DSI, the better an LLM is at that domain. If an LLM's DSI in a domain exceeds a threshold $\mathbf{T}$, we consider it a comfort domain. If an LLM's DSI in a domain is less than $-\mathbf{T}$, it is considered a strange domain. $\mathbf{T}$ is a hyper-parameter and is set to 10% in this paper. Practitioners can further tune this parameter.

Table 7 shows the DSIs of studied LLMs across domains. The comfort domains and strange domains are marked in bleu and red, respectively. We can see that gpt-4 has the most comfort domains. Especially in the Text Processing domain, among all LLMs, only gpt-4 successfully solves some programming tasks. However, gpt-4 performs worse than others in the Internet domain. Besides, we discover that StarCoder 2-15B unexpectedly performs well in the Database domain and even is comparable to gpt-4. The potential reason for comfort and strange domains is that the pre-training data mix of LLMs is different. For example, gpt-4's training data contains fewer repositories in the Internet domain, resulting in weak performance. These findings can help model trainers analyze the shortcomings of existing LLMs and build more powerful code LLMs.

## 4 Discussion

**Evaluation of auto-generated annotations.** We leverage an LLM (*i.e.,* gpt-4 in this paper) to annotate natural language requirements and domain labels for functions. To assess the quality of auto-generated annotations, We hire five developers to write requirements and domain labels for EvoCodeBench-2403. Then, we hire another five developers to compare annotations from gpt-4 and developers. All of these developers have at least 3 years of Python development experience. All developers are paid according to the relevant policies[3] (*i.e.,* $7.5 per hour). The details of human evaluation are in Appendix E.3.

Table 8: Human evaluation of auto-generated annotations.

| Annotator | Win / Tie / Lose Requirement | Win / Tie / Lose Domain | Cost (Time) | Cost (Money) |
|---|---|---|---|---|
| gpt-4-turbo-1106 | 30 / 236 / 9 | 3 / 268 / 4 | 1h9m | $3.11 |
| Human | 9 / 236 / 30 | 4 / 268 / 3 | 23h | $172.5 |

The evaluation results are shown in Table 8. The Cohen's Kappa coefficient between all evaluators is 0.9. The *Tie* means both requirements are satisfying. We can see that gpt-4 produces high-quality annotations comparable to human-written annotations in most cases (*e.g.,* requirement: 96.7% = (30+236)/275, domain labels: 98.5% = (3+268)/275). We also inspect failed cases of gpt-4 and summarize two main reasons. First, gpt-4 may miss some

---

[3]https://www.worker.gov/

details (*e.g.,* hyper-parameters) that are necessary for requirements. Second, gpt-4 may be mistaken by specific APIs and output inaccurate domain labels. In the future, we will explore new techniques to solve this problem, *e.g.,* controllable text generation [4]. Besides, gpt-4 shows advantages in costs. As shown in Table 8, gpt-4 costs less time and money to annotate requirements. Thus, it is a feasible and efficient approach for us to use gpt-4 to annotate requirements for EvoCodeBench.

**Limitations.** There are two main limitations in EvoCodeBench. First, EvoCodeBench is a monolingual (*i.e.,* Python) benchmark and ignores other programming languages (*e.g.,* Java, C++). Because building repo-level benchmarks faces many language-specific challenges (*e.g.,* how to install execution environments, how to run test cases), we chose to start with Python, a mainstream programming language in existing benchmarks [28, 3, 7, 29]. We plan to support other programming languages in the future gradually. Second, the size of EvoCodeBench is currently smaller than some existing benchmarks. The reason is that EvoCodeBench-2403 only collects samples from recent repositories (*i.e.,* Oct. 2023 - Mar. 2024). In the future, we will continue to collect new samples from the latest repositories and expand the scale of EvoCodeBench.

## 5    Related Work

**Code Generation Benchmarks.** Nowadays, prevalent code generation benchmarks can be divided into two groups: ❶ Snippet-level benchmarks [3, 1, 11, 7]. They comprise human-written or competitive programming problems, which ask LLMs to generate standalone code snippets. ❷ Repo-level benchmarks [29, 16]. They ask LLMs to generate new programs based on requirements and contexts from current repositories. Compared to snippet-level benchmarks, repo-level benchmarks are more challenging and closer to real-world software development scenarios.

This paper proposes a new benchmark - EvoCodeBench, to alleviate two limitations of previous benchmarks (*i.e.,* data leakage and lack of domain-specific evaluations). We notice that some recent benchmarks focus on similar limitations. We further clarify the differences between EvoCodeBench and existing benchmarks.

**Data leakage.** LiveCodeBench [13] collects the latest competitive programming problems. EvoEval [27] leverages LLMs to mutate HumanEval and obtain new benchmarks. They are both snippet-level benchmarks, while EvoCodeBench is a more practical repo-level benchmark. The collection pipelines in LiveCodeBench and EvoEval can not be applied to repo-level benchmarks, which involve many new challenges (*e.g.,* repository selection, test construction, and requirement annotation). We fill this knowledge gap by building a new collection pipeline and release EvoCodeBench.

**Domain-specific evaluations.** Existing benchmarks typically fall into narrow domains (*e.g.,* PandasEval [30]) or lack domain labels (*e.g.,* DevEval [16]). ClassEval [7] contains 100 standalone programming tasks from seven domains. However, these domains are manually designed based on human experiences and may ignore important domains (*e.g.,* Internet and Multimedia). Besides, ClassEval ignores repo-level benchmarks and may be leaked in the future. In contrast, we consider the statistics of a mainstream open-source community and identify the top 10 popular programming domains. Besides, EvoCodeBench is free of data leakage and continually expands domains.

## 6    Conclusion and Future Work

We introduce EvoCodeBench, an evolving code generation benchmark. EvoCodeBench is designed to address two limitations (*i.e.,* data leakage and lack of domain-specific evaluations). EvoCodeBench is an evolving benchmark and will be dynamically updated every period (*e.g.,* six months), to avoid data leakage. This paper releases the first version - EvoCodeBench-2403, which contains 275 samples. Besides, we design a programming domain taxonomy consisting of ten popular domains and annotate samples with domain labels. We conduct extensive experiments on EvoCodeBench and reveal the actual abilities of LLMs in real-world repositories. We also evaluate LLMs in different domains and discover their comfort and strange domains. These insights can help practitioners evaluate LLMs comprehensively.

In the future, we will continuously release new versions of EvoCodeBench and extend EvoCodeBench into other programming languages (*e.g.,* Java and C++).

## Acknowledgements

This research is supported by the National Natural Science Foundation of China (Nos. 62192731, 62152730), the National Key R&D Program under Grant No. 2023YFB4503801, the National Natural Science Foundation of China (Nos. 62072007, 62192733, 61832009, 62192730), and the Major Program (JD) of Hubei Province (No.2023BAA024). Ge Li and Yongbing Li are the corresponding authors.

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

## Appendix

## Table of Contents

## A   Hosting, Licensing, and Maintenance

Our EvoCodeBench and experimental results (*e.g.,* code, prompts, and models' predictions) are available on the following platforms.

- GitHub: `https://github.com/seketeam/EvoCodeBench`
- HuggingFace: `https://huggingface.co/datasets/LJ0815/EvoCodeBench`
- Croissant metadata: `https://github.com/seketeam/EvoCodeBench/blob/main/croissant_metadata.json`

EvoCodeBench is available for download under a CC-4.0 license, and our code is available under a BSD 3-Clause license. We ensure the long-term availability and maintenance of the data by hosting it on the GitHub[4] platform.

## B   Author Responsibility Statement

The authors confirm that they bear all responsibility in case of any rights violation during the data collection or other work and will take appropriate action when needed, *e.g.,* to remove data with such issues. The authors also confirm the licenses provided with the data and code associated with this work.

## C   Datasheet for EvoCodeBench

Questions from Datasheet for Datasets (v8) [9].

### C.1   Motivation

**Q: For what purpose was the dataset created?**

Large Language Models (LLMs) have shown impressive abilities in code generation. This dataset was created to evaluate LLMs in code generation. It specifically fills in two knowledge gaps in previous benchmarks, *i.e.,* data leakage and lack of domain-specific evaluations. The former hurts the fairness of benchmarks, and the latter hinders practitioners from selecting superior LLMs for specific domains.

**Q: Who created the dataset (e.g., which team, research group) and on behalf of which entity (e.g., company, institution, organization)?**

This dataset's authors are from the School of Computer Science at Peking University and the Conversational AI team at Alibaba DAMO Academy.

**Q: Who funded the creation of the dataset?**

This dataset is funded by Peking University and the Alibaba DAMO Academy.

---

[4]`https://github.com/`

### C.2  Composition

**Q: What do the instances that comprise the dataset represent (e.g., documents, photos, people, countries)?**

An instance in the dataset represents a unique programming task. Each instance consists of the following seven components. (1) Function Signature: The signature of the target code. (2) Requirement: An English description detailing the functionality of the target code. (3) Repository: The current repository contains hundreds of code files. (4) Reference Code: A developer-written implementation of the target code. This code may invoke dependencies defined in the current repository. (5) Reference Dependency: The dependencies invoked in the reference code include intra-class, intra-file, and cross-file dependencies. (6) Domain Label: The domain of the target code. (7) Test Cases: Test cases are used to check the functional correctness of the code.

**Q: How many instances are there in total (of each type, if appropriate)?**

Our dataset is evolving and will be dynamically updated every period (*e.g.,* six months). In this paper, we release its first version - EvoCodeBench-2403, which contains 275 instances.

**Q: Does the dataset contain all possible instances or is it a sample (not necessarily random) of instances from a larger set?**

Our dataset - EvoCodeBench is evolving and contains a series of versions. In this paper, we release its first version - EvoCodeBench-2403. In the future, we will release new versions, *e.g.,* EvoCodeBench-2409. The data formats of different versions are the same.

**Q: What data does each instance consist of?**

Figure 1 shows an instance in our dataset. Each instance consists of the following seven components. (1) Function Signature: The signature of the target code. (2) Requirement: An English description detailing the functionality of the target code. (3) Repository: The current repository contains hundreds of code files. (4) Reference Code: A developer-written implementation of the target code. This code may invoke dependencies defined in the current repository. (5) Reference Dependency: The dependencies invoked in the reference code include intra-class, intra-file, and cross-file dependencies. (6) Domain Label: The domain of the target code. (7) Test Cases: Test cases are used to check the functional correctness of the code.

**Q: Is any information missing from individual instances?**

No.

**Q: Are relationships between individual instances made explicit (e.g., users' movie ratings, social network links)?**

No.

**Q: Are there recommended data splits (e.g., training, development/validation, testing)?**

Our dataset is a benchmark for code generation, which only contains test data.

**Q: Are there any errors, sources of noise, or redundancies in the dataset?**

As discussed in Section 4, we leverage LLMs to annotate requirements and domain labels of instances automatically. We conduct a human evaluation to assess the auto-generated annotations. The evaluation results (Table 8) show that auto-generated annotations are comparable to human-written annotations in most instances (*i.e.,* 96.7% requirements and 98.5% domain labels). In a small number of instances, auto-generated annotations may not be exactly correct, *e.g.,* missing necessary details in requirements. We think that these noises have a slight impact on our datasets. In the future, we will explore more techniques to address these noises.

**Q: Is the dataset self-contained, or does it link to or otherwise rely on external resources (e.g., websites, tweets, other datasets)?**

It is self-contained.

**Q: Does the dataset contain data that might be considered confidential (e.g., data that is protected by legal privilege or by doctor-patient confidentiality, data that includes the content of individuals' non-public communications)?**

No.

**Q: Does the dataset contain data that, if viewed directly, might be offensive, insulting, threatening, or might otherwise cause anxiety?**

No.

**Q: Does the dataset relate to people?**

No.

### C.3 Collection Process

**Q: How was the data associated with each instance acquired?**

Our dataset is collected by a four-stage pipeline, which includes (1) Repo selection and function scraping, (2) Execution-based filtering, (3) Automatic annotations, and (4) Benchmark construction. The pipeline details are in Section 2.3.

**Q: What mechanisms or procedures were used to collect the data (e.g., hardware apparatus or sensor, manual human curation, software program, software API)?**

Our collection pipeline involves the following four existing software tools or APIs.

- GitHub APIs[5]. We use this API to crawl open-source repositories from GitHub.
- Pip[6]. We use this software tool to install required packages for each repository automatically.
- Pytest[7]. We use this software tool to execute test cases.
- OpenAI API[8]. Through this API, we invoke gpt-4 to generate requirements and domain labels for instances.

**Q: If the dataset is a sample from a larger set, what was the sampling strategy (e.g., deterministic, probabilistic with specific sampling probabilities)?**

N/A

**Q: Who was involved in the data collection process (e.g., students, crowdworkers, contractors) and how were they compensated (e.g., how much were crowdworkers paid)?**

N/A

**Q: Over what timeframe was the data collected?**

The first version - EvoCodeBench-2403 is collected from repositories that were created between October 2023 and March 2024.

**Q: Were any ethical review processes conducted (e.g., by an institutional review board)?**

Yes. Our EvoCodeBench is a code-related benchmark. It is collected from high-quality open-source repositories and excludes malicious or offensive repositories.

### C.4 Preprocessing/cleaning/labeling

**Q: Was any preprocessing/cleaning/labeling of the data done (e.g., discretization or bucketing, tokenization, part-of-speech tagging, SIFT feature extraction, removal of instances, processing of missing values)?**

Yes. We filter out instances satisfying the following criteria: empty functions, initialization functions, and functions without executable test cases. Besides, we use a static analysis parser and an LLM to annotate these instances. Specifically, we use a static analysis parser to extract function signatures, reference code, and reference dependencies. Then, we use an LLM (gpt-4 in this paper) to annotate requirements and domain labels.

---

[5] https://docs.github.com/en/rest?apiVersion=2022-11-28
[6] https://pip.pypa.io/en/stable/installation/
[7] https://docs.pytest.org/en/8.2.x/
[8] https://platform.openai.com/docs/introduction

**Q: Was the "raw" data saved in addition to the preprocessed/cleaned/labeled data (e.g., to support unanticipated future uses)?**

No. Our dataset's raw data contains many large-scale code repositories, which are not conducive to downloading the dataset. We release the URLs of these raw repositories to support unanticipated future uses.

**Q: Is the software used to preprocess/clean/label the instances available?**

Yes. The used source code is available in `https://github.com/seketeam/EvoCodeBench`.

## C.5 Uses

**Q: Has the dataset been used for any tasks already?**

Our dataset is designed for the code generation task. In this paper, we have evaluated eight popular code LLMs in this dataset.

**Q: Is there a repository that links to any or all papers or systems that use the dataset?**

No.

**Q: What (other) tasks could the dataset be used for?**

Besides code generation, our dataset can support the following code intelligence tasks, including code completion, test cases generation, and code summarization.

**Q: Is there anything about the composition of the dataset or the way it was collected and preprocessed/cleaned/labeled that might impact future uses?**

No.

**Q: Are there tasks for which the dataset should not be used?**

No.

## C.6 Distribution

**Q: Will the dataset be distributed to third parties outside of the entity (e.g., company, institution, organization) on behalf of which the dataset was created?**

No.

**Q: How will the dataset will be distributed (e.g., tarball on website, API, GitHub)?**

It will be publicly available for download on GitHub (`https://github.com/seketeam/EvoCodeBench`).

**Q: Will the dataset be distributed under a copyright or other intellectual property (IP) license, and/or under applicable terms of use (ToU)?**

CC-4.0.

**Q: Have any third parties imposed IP-based or other restrictions on the data associated with the instances?**

No.

**Q: Do any export controls or other regulatory restrictions apply to the dataset or to individual instances?**

No.

## C.7 Maintenance

**Q: Who is supporting/hosting/maintaining the dataset?**

The SEKE team from Peking University will continuously maintain this dataset.

**Q: How can the owner/curator/manager of the dataset be contacted (e.g., email address)?**

Figure 3: The prompt template for generating requirements with gpt-4.

Please contact the first author - Jia Li (email: lijia@stu.pku.edu.cn).

**Q: Will the dataset be updated (e.g., to correct labeling errors, add new instances, delete instances)?**

Yes. We will continuously update EvoCodeBench and release new versions every period (*e.g.,* six months).

**Q: Will older versions of the dataset continue to be supported/hosted/maintained?**

Yes, older versions will remain available on GitHub.

**Q: If others want to extend/augment/build on/contribute to the dataset, is there a mechanism for them to do so?**

Not officially, but our benchmark code is open source and pull requests are welcome.

# D   Collection Pipeline Details

## D.1   Details of Automatic Annotation

As stated in Section 2.3, we leverage an LLM to annotate requirements and domain labels for candidate functions.

Figure 3 and Figure 4 show the prompt templates for generating requirements and domain labels, respectively. The parts highlighted in yellow in the figures are placeholders. {example_code} and {example_requirement} are a human-written function and its requirements, respectively. We fill {input_code} with candidate functions and leverage gpt-4 to generate requirements. We use the greedy search and generate a requirement for each function. The settings for generating domain labels are similar.

Please analyze the given Python code and select its domain keyword from a candidate list. You should directly output the most relevant domain and do not generate other explanations.

Candidate domains and their explanations:
- Communications: It includes Email, Chat, Fax, File Sharing, and Telephony.
- Database: It includes Database Engines, Database Servers, and Database Management.
- Internet: It involves FTP, HTTP, and Web Services.
- Multimedia: It typically processes Audio, Graphics, Video, and 3D Objects.
- Scientific Engineering: It includes Artificial Intelligence, Machine Learning, and Scientific Computing.
- Security: It includes Cryptography, Digital Signatures, and Secure Communication.
- Software Development: It includes Build Tools, Compilers, Debuggers, and IDEs.
- System: It includes File Systems, Operating Systems, and System Administration.
- Text Processing: It includes Markup Languages, Regular Expressions, and Text Analysis.
- Utilities: It includes Compression, Configuration, Logging, and Testing.

- - - - - - - - - - - - - - - - - - - - - - - - - - - - - - - - - - - - - - - - - - - - - - - - - -

Input Code:
```Python
{input code}
```

Domain:

Figure 4: The prompt template for generating domain labels with gpt-4.

Table 9: The statistics of 25 repositories on EvoCodeBench-2403.

| Repository | Created | Stars | Py Files | Py Lines | Samples |
|---|---|---|---|---|---|
| Test-Agent | 2023-10-20 | 440 | 85 | 15278 | 1 |
| skfolio | 2023-12-14 | 813 | 158 | 33852 | 13 |
| camp_zipnerf | 2024-01-19 | 523 | 53 | 18973 | 54 |
| microagents | 2023-12-11 | 674 | 45 | 2918 | 18 |
| open-iris | 2023-12-09 | 161 | 140 | 13933 | 14 |
| litdata | 2024-02-15 | 114 | 56 | 11713 | 59 |
| nlm-ingestor | 2024-01-17 | 643 | 56 | 16674 | 4 |
| AutoRAG | 2024-01-10 | 259 | 115 | 7735 | 13 |
| XAgent | 2023-10-16 | 7054 | 148 | 17623 | 3 |
| tanuki_py | 2023-10-16 | 606 | 108 | 10146 | 9 |
| UHGEval | 2023-11-06 | 148 | 34 | 2938 | 3 |
| Generalizable-BEV | 2023-10-30 | 136 | 570 | 132407 | 8 |
| EasyVolcap | 2023-12-07 | 442 | 308 | 51723 | 20 |
| UniRef | 2023-12-22 | 208 | 382 | 70042 | 23 |
| contrastors | 2024-01-30 | 346 | 62 | 13774 | 1 |
| gaussian-splatting-lightning | 2023-10-06 | 168 | 76 | 9935 | 1 |
| scepter | 2023-12-21 | 190 | 244 | 41519 | 1 |
| microsearch | 2024-02-05 | 336 | 5 | 231 | 2 |
| ollama-python | 2023-12-09 | 898 | 13 | 2089 | 12 |
| Python-Type-Challenges | 2023-10-23 | 343 | 121 | 3208 | 1 |
| stable-fast | 2023-10-17 | 871 | 82 | 11948 | 2 |
| stable-diffusion-webui-forge | 2024-01-14 | 2537 | 1112 | 210946 | 3 |
| openlogprobs | 2023-11-22 | 174 | 6 | 524 | 1 |
| searcharray | 2023-11-03 | 133 | 25 | 4217 | 6 |
| deluder | 2023-12-01 | 115 | 34 | 1894 | 3 |

## D.2 Repositories in EvoCodeBench-2403

Table 9 shows the statistics of 25 repositories in EvoCodeBench-2403.

# E Experimental Details

## E.1 Base LLMs

In this paper, we select five popular LLMs as base LLMs and evaluate them on EvoCodeBench-2403. The details of these LLMs are described as follows.

- **gpt-4** [21], released by OpenAI on March 14, 2023, marks another milestone in the field of natural language processing. gpt-4 demonstrates superior performance compared to previous gpt models [2]. In our experiments, we use the version - gpt-4-1106. Its training data up to April 2023. It continues the auto-regressive prediction of the next token training objective inherited from the GPT series models. It incorporates reinforcement learning with human feedback (RLHF) and red-teaming [8] techniques. However, the pre-training data scope and scale, model size, and parameters remain closed-source at present.

- **gpt-3.5-turbo** [20] is an improved gpt-3 model enhanced by a three-stage reinforcement learning with human feedback (RLHF) algorithm. Apart from improving instruction-following capabilities, the RLHF algorithm proves highly effective in mitigating the generation of harmful or toxic content, which is crucial for the practical deployment of LLMs in security-sensitive contexts. we utilized the released versions of gpt-3.5, namely gpt-3.5-turbo-1106, with training data up to September 2021. However, similar to gpt-4, the training details, training data, and model weights are currently closed-source.

- **CodeLLaMa** [26], based on the LLama2 architecture by Meta-AI[9], was fine-tuned and open-sourced by the company on August 25, 2023, with versions of 7B, 13B, and 34B. A 70B version was released on January 30, 2024 [26]. CodeLLama is primarily trained on nearly deduplicated publicly available code datasets. The first three models were trained on 500 billion tokenized code, while the latest 70B model was trained on 1T tokens. Similar to the LLaMa series, CodeLLaMa also follows a decoder-only architecture. We evaluated CodeLLaMa-Python-{7B, 13B} upon our EvoCodeBench.

- **DeepSeek Coder** [10] is a large language model for programming tasks released by DeepSeek-AI[10] in November 2, 2023. DeepSeek Coder consists of a series of code language models, each trained from scratch on 2T tokens, containing 87% code and 13% natural language. DeepSeek Coder provides code models with 1.3B, 6.7B and 33B parameter sizes. In terms of model architecture, each model integrates a decoder-only Transformer, incorporating Rotary Position Embedding and FlashAttention v2. We evaluated DeepSeek Coder-{6.7B, 33B} on our EvoCodeBench.

- **StarCoder 2** [19] was released by BigCode[11] on December 8, 2023 with 3 different parameters, 3B, 7B and 15B. StarCoder2 is trained on The Stack v2, a new large-scale, high-quality code dataset. All models were trained using Grouped Query Attention, a contextual window of 16,384 tokens with a sliding window attention of 4,096 tokens, using the Fill-in-the-Middle objective. Following DeepseekCoder [10] and Code LLaMA [26], StarCoder2 use Rotary Positional Encodings. We evaluated StarCoder2-{7B, 15B} on our EvoCodeBench, which was trained on over 3.5 trillion tokens in 17 programming languages from Stack v2.

## E.2 Prompt Templates

The prompt templates used for instruction-tuning models (*i.e.,* gpt-4 and gpt-3.5) are shown in Figure 5, 6, 7, and 8. {function name}, {contexts above}, {contexts below}, {signature}, and {requirement} are placeholders.

For other standard language models, the prompt templates are shown as follows:

- Without context: [signature; requirement]

- Local file (completion): [context_above; signature; requirement]

- Local file (infilling): [prefix_id; context_above; signature; requirement; suffix_id; context_below; middle_id]

Where [;] denotes the concatenation operation of strings. {prefix_id}, {suffix_id}, {middle_id} are special tokens used in code infilling. For different LLMs, we reuse their official special tokens to make prompts.

---

[9]https://ai.meta.com/

[10]https://www.deepseek.com/

[11]https://www.bigcode-project.org/

```
Please complete the {function name} function in the given Python
code.
----------------------------------------------------------
Input Code:
```Python
{signature}
{requirement}
```

Completed Code:
```

Figure 5: The prompt template in the without context setting.

```
Please complete the {function name} function based on the
contexts above the function.
----------------------------------------------------------
The contexts above the function are:
```Python
{contexts above}
```
----------------------------------------------------------
The code to be completed is:
```Python
{signature}
{requirement}
```

Completed code:
```

Figure 6: The prompt template in the local file (completion) setting.

### E.3 Details of Human Evaluation

Figure 9 and Figure 10 show the questionnaire templates for evaluating requirements and domain
labels, respectively. The parts highlighted in yellow in the figures are placeholders. Taking Figure 9
as an example, we randomly arrange auto-generated requirements and human-written requirements
and then fill them into the placeholders. The setup for evaluating domain labels is similar.

Please complete the *{function name}* function in the middle of a file.

----------------------------------------------------------------

The contexts above the function are:
```Python
{contexts above}
```

The contexts below the function are:
```Python
{contexts below}
```

The code to be completed is:
```Python
{signature}
{requirement}
```

Completed code:

Figure 7: The prompt template in the local file (infilling) setting.

Please complete the *{function name}* function based on some functions with similar names.

----------------------------------------------------------------

The functions with similar names are:
```Python
{similar functions}
```

----------------------------------------------------------------

The code to be completed is:
```Python
{signature}
{requirement}
```

Completed code:

Figure 8: The prompt template in the similar function setting.

Based on the code, please choose which requirement is better based on the following two perspectives:
(1) Completeness (the requirements cover the main purpose of the code and necessary details); (2)
Clarity (requirements are clear and user-friendly). If you think both requirements are good, select "Tie".

Code:
*{input code}*

Requirement #1:
*{requirement_1}*

Requirement #2:
*{requirement_2}*

--------------------------------------------------------------------------------

☐ Requirement #1  ☐ Requirement #2  ☐ Tie

Figure 9: The questionnaire template for evaluating requirements.

Based on the code, please choose which domain label is better. If you think both domain labels are
reasonable, select "Tie".

Code:
*{input code}*

Domain Label #1:
*{requirement_1}*

Domain Label #2:
*{requirement_2}*

--------------------------------------------------------------------------------

☐ Domain Label #1  ☐ Domain Label #2  ☐ Tie

Figure 10: The questionnaire template for evaluating domain labels.

