# OpenReview forum: "EvoCodeBench: An Evolving Code Generation Benchmark with Domain-Specific Evaluations"
_NeurIPS.cc/2024/Datasets_and_Benchmarks_Track — NeurIPS 2024 Track Datasets and Benchmarks Poster_

### Official Review · Reviewer_CzHa · 2024-07-20
**EvoCodeBench: An Evolving Code Generation Benchmark with Domain-Specific Evaluations**

**Rating:** 5
**Confidence:** 3
**Correctness:** Needs to illustrate more.
**Clarity:** Needs to strengthen.

**Review:**

The paper has some pros and cons.

Pros:
1. EvoCodeBench is designed to address two limitations (i.e., data leakage and lack of domain-specific evaluations).
2. EvoCodeBench is an evolving benchmark and will be dynamically updated every period (e.g., six months), to avoid data leakage.
3. It designs a programming domain taxonomy consisting of ten popular domains and annotate samples with domain labels.

Cons:
 1. The paper lacks a clear illustration on the underlying technical aspects of the new designing.
 2. The originality of this paper shall be explained in detail.
 3. The paper shall strengthen the illustration the significance of this work.

**Strengths:**

The strengths of this paper include:

EvoCodeBench is designed to address two limitations (i.e., data leakage and lack of domain-specific evaluations).  These are important aspects of the AI research community.

**Additional Feedback:**

NA

**Documentation:**

Yes

**Opportunities For Improvement:**

1. The paper lacks a clear illustration on the underlying technical aspects of the new designing.
 2. The originality of this paper shall be explained in detail.
 3. The paper shall strengthen the illustration the significance of this work.

**Relation To Prior Work:**

Yes

**Summary And Contributions:**

In this paper, the authors proposed a new benchmark, EvoCodeBench, which has the following advances: 1. dynamically updated every period (e.g., 6 months) to avoid data leakage; 2. a programming domain taxonomy consisting of 10 popular domains; 3.  compute the Domain-Specific Improvement (DSI) and define LLMs’ comfort and strange domains.  EvoCodeBench has an advantage in data quality: EvoCodeBench
is collected from high-quality open-source repositories and it offers comprehensive annotations.

---

> ### Author Rebuttal · Authors · 2024-08-16
>
> ### **1. Response to C1 (the underlying technical aspects)**
>
> Thank you for the suggestion. Our paper proposes a new code generation benchmark named EvoCodeBench. Its technical aspects mainly include a benchmark collection pipeline and a large-scale evaluation. The related contents are available in Sections 2 and 3.
>
> To provide a more comprehensive response, could you please explain which technical aspect you have concerns about? Thank you!
>
> ### **2. Response to C2 (The originality of this paper...)**
>
> Thank you for the comment. Our EvoCodeBench aims to address two limitations of previous benchmarks: data leakage and lack of domain-specific evaluations). As stated in Section 5, we state our originality by comparing EvoCodeBench to related works. (1) Data Leakage. Previous studies limit snippet-level benchmarks, while EvoCodeBench is a practical repo-level benchmark. Our EvoCodeBench is closer to real-world repositories and evaluates the coding abilities of LLMs in the software development  (2) Domain-specific Evaluations. Previous studies typically fall into narrow domains or lack domain labels. EvoCodeBench covers PyPI's top 10 popular programming domains and offers fine-grained domain labels. We also conduct domain-specific evaluations of 10 LLMs in Section 3.4.
>
> We hope our response addresses your concerns. If you have any further questions, please let us know.
>
> ### **3. Response to C3 (The illustration of the significance of this work..)**
>
> Thank you for the suggestion. A high-quality code generation evaluation benchmark can effectively distinguish superior and inferior models and facilitate the development of code generation. Although many benchmarks have been proposed, they have two limitations:
>
> - Data leakage (aka data contamination). It means that test data is included in the training data. The trained models perform much better on leaked benchmarks than on real-world tasks. Because the training data of LLMs contains almost all open-source code repositories, existing benchmarks probably have data leakages.
>
> - Lack of domain-specific evaluation. Programming is highly domain-specific. Developers typically focus on specific domains (e.g., Database) and are concerned about the performance of LLMs in specific domains. However, existing benchmarks lack domain labels or fall into narrow domains. Besides, they ignore domain-specific evaluations and analyses. Thus, the performance of LLMs across domains is still unclear.
>
> The main contributions of this paper are to address the above limitations:
>
> - Evolving data. To avoid data leakages, EvoCodeBench is an evolving benchmark that will be dynamically updated every period (i.e., 6 months in this paper).
>
> - Domain labels and domain-specific evaluations. We collect the domain distribution of a popular open-source community - PyPI, and summarize a domain taxonomy. Based on domain taxonomy, we annotate each sample in EvoCodeBench with a domain label. Then, we conduct domain-specific evaluations and propose Domain-Specific Improvement (DSI), which reflects the position of an LLM in specific domains.
>
> **Implication To Practitioners.** Compared to existing benchmarks, EvoCodeBench offers a more fair and fine-grained evaluation platform. It avoids the impact of data leakages and fairly ranks different LLMs in code generation. Besides, it provides fine-grained domain labels and helps practitioners select superior models in specific domains. Nowadays, EvoCodeBench is used by academic researchers [1] and IT companies (e.g., Alibaba and ByteDance).
>
> [1] Liu, Xiangyan, et al. "CodexGraph: Bridging Large Language Models and Code Repositories via Code Graph Databases." arXiv preprint arXiv:2408.03910 (2024).

---

> ### Author Rebuttal · Authors · 2024-08-21
>
> Dear Reviewer CzHa,
>
> Thank you again for reviewing our paper and for the valuable feedback. We have made every effort to address your concerns. As the rebuttal period is coming to an end, we are eager to know any additional comments or questions you may have. Thank you again for your time!
>
> Sincerely,
>
> Authors

---

### Official Review · Reviewer_LgtM · 2024-07-22
**Periodically Updated Repository-level Code Generation Benchmark with Domain Labels**

**Rating:** 6
**Confidence:** 4
**Correctness:** 1. The equation for Pass@k is wrong. …
**Clarity:** Yes

**Review:**

Although there are several typos and mistakes, the paper is well-structured and easy to understand. The authors explain the way the benchmark is constructed and conduct abundant experiments on the benchmark. However, some parts of the explanation of data collection are omitted and the formal definition for the suggested metric, DSI, is missing. The proposed methods to mitigate the data leakage problem and the suggested programming domain taxonomy are novel. The authors provide the experimental results showing that the considered problem is meaningful and significant. The claim is convincing since the data leakage problem in commercial or open-source LLMs is getting bigger and bigger.

Pros :
1. The provided benchmark is data leakage-free.
2. The distribution of the provided benchmark is similar to that of real-world Github repositories.
3. The provided benchmark makes it possible to evaluate models in a specific domain.


Cons :
1. The explanation about constructing the benchmark is not enough.
2. The proposed metric is not formally defined.
3. The number of data in a specific domain is not enough for evaluation.
4. Once the benchmark is updated, models need to be evaluated again to be compared with those developed in the future.

**Strengths:**

The contribution of this work seems nice since it can help to compare several LLMs without concerning the data leakage issue. It is clear that the concept of a periodically updated benchmark can mitigate data leakage because a model cannot be trained on the data from the repositories created after the training.

The authors label the data in the benchmark based on the statistics of PyPI and the domain taxonomy is convincing and useful. Also, the sampling based on the real-world distribution makes the benchmark more realistic. Models can be evaluated on a specific domain in the benchmark and their relative performance can be evaluated on the suggested metric.

**Additional Feedback:**

Typos and Suggestions
+ In line 95, Recallk -> Recall@k
+ In line 101, a citation for the previous work is needed.
+ In line 176, the new line is improperly used.
+ In line 267, got-4 -> gpt-4
+ In Table 4, bold and underline are misused in local file infilling Pass@5 & Pass@10 and local file completion Recall@10.
+ In Table 6, I think the word “bleu” in the title should be “blue.”

Questions
+ The contribution of this work is meaningful while it is periodically updated. I would like to know how long exactly you plan to provide support. Additionally, I would like to know the specific plan for EvoCodeBench-2409, including the support of programming languages other than Python and the expected scale of the benchmark.

**Documentation:**

Yes

**Ethics:**

They need to anonymize the codes from real-world repositories since they may contain some personal information like a phone number, URL and email in the comments.

**Limitations:**

Yes

**Opportunities For Improvement:**

A figure explaining the overall workflow of constructing the benchmark is needed.
It would be better to conduct additional experiments for data leakage detection on the DevEval benchmark, which is the prior work of the authors with the same task, repository-level code generation.
The statistics of the domain distribution in PyPI should be addressed in the paper to be compared with those of the benchmark.
Some minor typos or suggestions are in the additional feedback.

Codes in some repositories might be the same or similar to codes in other repositories. Thus, the data leakage problem has not been completely resolved and it would be better to clarify how this problem can be addressed. One possible suggestion is to use databases or collections of codes, which determine whether a code in the chosen repository is unique or not.

**Relation To Prior Work:**

Yes

**Summary And Contributions:**

The paper considers the problem of data leakage in code datasets and releases a new benchmark, EvoCodeBench-2403, for the repository-level code generation. The authors promise that the benchmark will be periodically updated by crawling codes from the latest Github repositories to avoid data leakage, which is the main difference from their prior work. The proposed benchmark has a programming domain taxonomy with 10 popular domains in PyPI. The authors evaluate 8 popular LLMs, including GPT-4, on the proposed benchmark and report Pass@k and Recall@k. The models have a hard time with code generation when compared to the prior work, which means there is indeed data leakage. The authors also propose a new metric that measures how well an LLM performs in a specific domain compared to other LLMs.

---

> ### Author Rebuttal · Authors · 2024-08-16
>
> ### **1. Response to C1 (More explanations about construction)**
>
> Thank you for the suggestion. Figure 1 in the uploaded PDF file shows our collection pipeline. As stated in Section 2.3, our collection pipeline consists of four stages:
>
> - **Repository Selection and Function Scraping.** We crawl high-quality repositories from GitHub and extract candidate functions from them. Trivial functions (e.g., empty functions) are excluded.
>
> - **Execution-based Filtering.** For each candidate function, we extract its test cases from the repository, construct the running environment, and execute the test cases. Candidate functions without successful test cases will be excluded.
>
> - **Automatic Annotations.** We use a static analysis parser for each candidate function to extract its signature, function body, and invoked dependencies. Then, we use an LLM (GPT-4-Turbo in this paper) to generate requirements and domain labels for candidate functions.
>
> - **Benchmark Construction.** We randomly select a few candidate functions to construct EvoCodeBench and ensure its code distribution is consistent with 500 real-world repositories.
>
> ### **2. Response to C2 (Metric is not formally defined)**
>
> (1) Sorry, the equation for computing Pass@k contains a typo. The corrected equation is in the uploaded PDF file (Figure 2). Besides, the code implementation of Pass@k in our repository is correct. Thus, the evaluation results are trustworthy.
>
> (2) As stated in Line 235, the DSI refers to the average relative improvement of Pass@1 of an LLM in a domain compared to other LLMs. Suppose we evaluate $N$ LLMs on a specific domain, and their Pass@1 scores are represented as $P$. Then, the DSI (%) of $i$-th LLM in this domain is computed by as:
> \begin{equation}
> DSI_i = \frac{1}{N-1}\sum_j \frac{P_i-P_j}{P_i} * 100 \quad (i \neq j)
> \end{equation}
>
> Where $P_i$ and $P_j$ are Pass@1 scores of $i$-th and $j$-th LLMs, respectively.
>
> ### **3. Response to C3 (Limited data in a specific domain)**
>
> **EvoCodeBench is an evolving benchmark that will be continually expanded in the future.** EvoCodeBench-2403 only considers repositories created from 2023-10 to 2024-03. In the future, we will collect new samples from the latest repositories and expand the data sizes of different domains. For example, we plan to release the second version, EvoCodeBench-2409, in Oct. 2024, which covers repositories created from 2023-04 to 2024-09. EvoCodeBench-2409 contains over 600 samples and covers 10 programming domains. Each domain will include at least 50 samples.
>
> **To improve the reliability of the evaluation, we evaluated LLMs in domains including more than 10 samples.** As stated in Line 225, the results in Tables 5 and 6 only consider seven domains with more than 10 samples and leave other domains for future work. Compared with the simple programming problems in existing benchmarks, the samples in EvoCodeBench are more challenging and can better evaluate the coding ability of LLMs. For example, the Database domain consists of 18 samples from 5 repositories. LLMs must understand detailed requirements (average: 142.3 tokens) and a long context (up to thousands of tokens on average) to solve a sample.  Although the data sizes of several domains are temporarily limited, EvoCodeBench-2403 can effectively rank LLMs.
>
> ### **4. Response to C4 (Models need to be evaluated again)**
>
> Thank you for the comment. As EvoCodeBench is updated, practitioners need to evaluate the LLMs in the updated version. We take the following steps to facilitate practitioners’ evaluation.
>
> - We have released all source code and scripts to automate the evaluation. Our scripts automatically generate prompts in different experimental settings and evaluate LLMs' completions. Thus, the whole evaluation process does not require human intervention.
>
> - The data structures of different versions of EvoCodeBench are consistent. Practitioners can reuse the source code and scripts to automate the evaluation in the future.
>
> - Our experiments use greedy search to sample a program to calculate Pass@1 and DSI scores. The cost of inference is relatively slight.
>
> ### **5. Response to C5 (data leakage detection on DevEval)**
>
> Following the settings in Section 3.2, we leverage CCD to detect DevEval's data leakage ratios. Table 1 in the uploaded PDF file shows the detection results. DevEval has a slight data leak and may be further leaked in the future. Thus, we propose EvoCodeBench to address the data leakage problem.
>
> ### **6. Response to C6 (Domain distributions in PyPI)**
>
> Table 2 in the uploaded PDF file shows the domain distribution in PyPI. We show the top 10 domains which our EvoCodeBench-2403 covers. We will introduce more samples in these domains in future versions and ensure a real domain distribution. Because other benchmarks do not provide domain labels, we can not directly compare their domain distributions with those in PyPI.
>
> ### **7. Response to C7 (Similar programs between repos)**
>
> Thank you for the suggestion. You are professional. Different repositories may contain similar programs, e.g., utility functions. To avoid data leakage caused by these similar programs, we plan to add a new step in data collection - code deduplication. We will construct a large-scale codebase storing many high-quality repositories. For each candidate function, we use Jaccard distance to retrieve similar functions from the codebase. We ignore this candidate function if the maximum Jaccard distance exceeds a threshold (e.g., 0.85). In this way, we ensure the programs to be generated in EvoCodeBench are not leaked in previous repositories.
>
> ### **8. Response to C8 (Personal Information) and C9 (Plan for EvoCodeBench)**
>
> Thank you for the suggestions. Our responses are in the **Response to C3** and **Response to C4** in the above **Response to All Reviewers**.
>
> ### **9. Response to C10 (Typos)**
>
> Thank you for the suggestion. We will revise these typos in our paper.

---

> ### Author Rebuttal · Authors · 2024-08-21
>
> Dear Reviewer LgtM,
>
> Thank you again for reviewing our paper and for the valuable feedback. We have made every effort to address your concerns. As the rebuttal period is coming to an end, we are eager to know any additional comments or questions you may have. Thank you again for your time!
>
> Sincerely,
>
> Authors

---

> > ### Comment · Reviewer_LgtM · 2024-08-28
> >
> > I appreciate your detailed response. I increase my evaluation accordingly. I hope that you will include additional information and the experimental results reported during the rebuttal in the revised version.

---

> > > ### Author Response · Authors · 2024-08-28
> > >
> > > Thank you for your invaluable comments. We will include additional information and the experimental results reported in the revised version.

---

### Official Review · Reviewer_7mRR · 2024-07-23
**Review for EvoCodeBench: An Evolving Code Generation Benchmark with Domain-Specific Evaluations**

**Rating:** 6
**Confidence:** 3

**Review:**

**Quality:**
EvoCodeBench provides a high-quality dataset with comprehensive annotations, including function signatures, natural language requirements, reference code, and dependencies. The dataset is collected using a robust and automated pipeline, ensuring relevance and minimizing data leakage by regularly incorporating new repositories.

**Clarity:**
The paper is well-structured and clearly explains the benchmark's methodology, data collection process, and evaluation metrics. Detailed figures and tables enhance understanding, and the documentation supports reproducibility and transparency.

**Originality:**
EvoCodeBench introduces an innovative approach by dynamically updating the dataset to address data leakage, a significant issue in LLM evaluation. The inclusion of domain-specific evaluations and diverse programming contexts further sets it apart from existing benchmarks.

**Significance:**
The benchmark's evolving nature and comprehensive coverage of multiple programming domains make it a valuable resource for evaluating LLMs in real-world scenarios. Its design addresses modern challenges in LLM evaluation, such as data leakage, ensuring that it remains relevant and useful for ongoing research and development.

**Strengths:**

1. **More Useful in Benchmarking LLMs:** EvoCodeBench is particularly effective for benchmarking LLMs because it highlights significant performance drops compared to previous benchmarks. The Pass@1 scores of gpt-4 and other models are substantially lower on EvoCodeBench, demonstrating that it presents more challenging and realistic scenarios. This makes EvoCodeBench a critical tool for revealing the true capabilities and limitations of LLMs in code generation tasks.

2. **Providing Domain-Specific Evaluation:** EvoCodeBench offers detailed domain-specific evaluations, covering diverse programming areas such as Database, System, Software Development, and more. This approach allows for a nuanced understanding of LLM performance across different domains. By identifying comfort and strange domains for each LLM, EvoCodeBench helps pinpoint areas where models excel or struggle, providing valuable insights for targeted improvements and practical applications.

3. **Rigorous Collection Process:** The benchmark employs a rigorous data collection and annotation process, ensuring high-quality and relevant samples. The pipeline includes repository selection, function scraping, execution-based filtering, and automatic annotations. This comprehensive and systematic approach ensures that the dataset is robust and reflective of real-world coding practices, enhancing its reliability and utility for evaluating LLMs.

**Additional Feedback:**

No additional feedback.

**Clarity:**

Yes, the paper is well written. It is clearly structured, with detailed explanations of the benchmark's methodology, data collection process, and evaluation metrics. The use of figures and tables enhances understanding, and the comprehensive documentation supports reproducibility and transparency.

**Correctness:**

Yes, the claims made in the submission appear to be correct. The dataset is constructed in a sound way, employing a rigorous and automated collection and annotation process. As a benchmark, the evaluation methods and experiment design are appropriate and performed correctly, using metrics like Pass@k and Recall@k to assess LLM performance in a comprehensive and nuanced manner.

**Documentation:**

Yes, there is sufficient detail on data collection, organization, availability, and maintenance. The dataset includes comprehensive documentation, intended uses, and a URL (https://github.com/seketeam/EvoCodeBench) for access. The authors also provide a hosting, licensing, and maintenance plan. For the benchmark, the detailed methodology and evaluation metrics support reproducibility.

**Ethics:**

No, I do not suspect there are any ethical concerns with the submission that warrant further discussion or review.

**Limitations:**

The authors have addressed some limitations of their work, particularly the issue of data leakage, by regularly updating the dataset. However, they could further elaborate on the potential negative societal impacts, such as the misuse of code generation for malicious purposes.

Constructive suggestions for improvement include discussing ethical guidelines for the responsible use of the dataset and exploring mechanisms to prevent the generation of harmful code. Additionally, being upfront about the limitations related to the dataset size and monolingual focus would further enhance transparency and provide a clearer roadmap for future enhancements.

**Opportunities For Improvement:**

1. **Questionable Scale of the Current Data Amount:** Although EvoCodeBench provides high-quality and well-annotated samples, the current dataset size may be insufficient for comprehensive evaluations. With only 275 samples in the first version, expanding the dataset to include more samples across various domains would enhance its robustness and generalizability, providing a more extensive foundation for evaluating LLMs.

2. **Lack of Cross-Domain Evaluation:** While the benchmark excels in domain-specific evaluations, it lacks a cross-domain evaluation framework that assesses LLM performance across multiple domains simultaneously. Incorporating such an evaluation would offer insights into the models' versatility and ability to generalize across different programming contexts, revealing potential strengths and weaknesses that might not be apparent in single-domain evaluations.

3. **Monolingual Focus:** The current version of EvoCodeBench focuses exclusively on Python, which limits its applicability to other programming languages. Expanding the benchmark to include additional languages like Java, C++, and JavaScript would make it more versatile and relevant to a broader range of applications, providing a more comprehensive evaluation of LLM capabilities across different coding environments.

**Relation To Prior Work:**

Yes, the paper clearly discusses how this work differs from previous contributions. It highlights the innovative approach of dynamically updating the dataset to minimize data leakage and the inclusion of domain-specific evaluations, which set EvoCodeBench apart from existing benchmarks.

**Summary And Contributions:**

The paper introduces **EvoCodeBench**, an evolving benchmark designed to evaluate LLMs in code generation. EvoCodeBench addresses two major limitations of existing benchmarks: data leakage and lack of domain-specific evaluation. The benchmark dynamically updates every six months to avoid data leakage and includes a domain taxonomy for domain-specific evaluations. The first version, EvoCodeBench-2403, contains 275 samples from 25 repositories and provides comprehensive annotations including natural language requirements, reference code, and test cases.

---

> ### Author Rebuttal · Authors · 2024-08-16
>
> ### **1. Response to C1 (Questionable scale of the current data amount...)**
>
> Thank you for the suggestion. Our response is in the **Response to C1** in the above **Response to All Reviewers**.
>
> ### **2. Response to C2 (Lack of Cross-Domain Evaluation...)**
>
> Thank you for the comment. There may be some misunderstanding. In Section 3.3, we evaluate 10 LLMs on samples from all domains and report their Pass@k and Recall@k in Table 4. The results reflect the comprehensive performance of LLMs across ten domains. GPT-4 achieves the best results among 10 studied LLMs. We also analyze successful and failed cases and discuss future directions. In Section 3.4, we evaluate the performance of LLMs in individual domains.
>
> ### **3. Response to C3 (Monolingual Focus...)**
>
> Thank you for the suggestion. Our response is in the **Response to C2** in the above **Response to All Reviewers**.
>
> ### **4. Response to C4 (Misuse of code generation...)**
>
> Thank you for the suggestion. You are right. It is important to prevent the misuse of code generation for malicious purposes. This paper proposes a new code generation benchmark, which aims to evaluate the capability of LLMs in generating normal programs. **As stated in Section 2.3, we only select non-malicious and high-quality repositories.** We will also ensure that future versions of EvoCodeBench do not contain malicious programs. Therefore, we believe that EvoCodeBench does not lead to the misuse of code generation.

---

> ### Author Rebuttal · Authors · 2024-08-21
>
> Dear Reviewer 7mRR,
>
> Thank you again for reviewing our paper and for the valuable feedback. We have made every effort to address your concerns. As the rebuttal period is coming to an end, we are eager to know any additional comments or questions you may have. Thank you again for your time!
>
> Sincerely,
>
> Authors

---

### Official Review · Reviewer_ci62 · 2024-07-25
**EvoCodeBench: An Evolving Code Generation Benchmark with Domain-Specific Evaluations**

**Rating:** 7
**Confidence:** 2

**Review:**

EvoCodeBench presents a valuable contribution to the field of code generation evaluation. The authors have identified and effectively addressed critical limitations in existing benchmarks.

Pros:
1. Evolving nature mitigates data leakage issues
2. Domain taxonomy enables nuanced performance analysis across programming areas
3. Aligns well with real-world repository characteristics, enhancing practical relevance
4. Comprehensive annotations and evaluation metrics (Pass@k and Recall@k)
5. Reveals interesting insights on LLM performance across different domains

Cons:
1. Current version (EvoCodeBench-2403) has a relatively small sample size
2. Limited to Python, restricting applicability to other programming languages
3. Potential biases in auto-generated annotations, despite human evaluation

The paper is well-structured and presents a novel approach to benchmark construction. The insights provided by EvoCodeBench are likely to be valuable for both researchers and practitioners in the field of code generation and LLM evaluation.

**Strengths:**

1. Novel approach to addressing data leakage through regular benchmark updates
2. Introduction of a domain taxonomy for more granular insights into LLM performance
3. Alignment with real-world repository characteristics, enhancing practical relevance
4. Comprehensive annotations and evaluation metrics for thorough assessment
5. Revealing insights into LLM performance across various programming domains

**Additional Feedback:**

(---)

**Clarity:**

The paper is generally well-written and logically structured. Clear explanations of methodology, benchmark construction, and evaluation processes are provided. Figures and tables enhance the presentation. Some technical sections, particularly those describing the benchmark construction pipeline, could benefit from additional clarification or examples to improve readability for a broader audience.

**Correctness:**

The claims made in the submission appear correct, and the methodology is sound. The benchmark construction process is well-described, and the evaluation methods are appropriate. Detailed information about the experimental setup and evaluated LLMs enhances reproducibility. The use of established metrics alongside the novel Domain-Specific Improvement (DSI) metric is suitable for the task.

**Documentation:**

Sufficient detail is provided on data collection, organization, and ethical considerations. The benchmark collection pipeline, including repository selection criteria and annotation processes, is described. The availability of the benchmark and associated code is discussed, enhancing reproducibility. The inclusion of a datasheet for the dataset in the appendix further strengthens the documentation.

**Ethics:**

No significant ethical concerns are apparent. The authors state they only consider code repositories with open-source licenses and exclude malicious code repositories. A human evaluation was conducted to assess the quality of auto-generated annotations, addressing potential concerns about using LLMs in the annotation process. The evolving nature of the benchmark and its focus on avoiding data leakage demonstrate a commitment to fairness and accuracy in evaluating LLMs. Ongoing attention to ethical considerations will be important as the benchmark grows and potentially incorporates more languages, particularly regarding data privacy, copyright, and consent for the use of open-source code in benchmark construction.

**Limitations:**

The authors adequately address the limitations of their work. They acknowledge the current focus on Python and the relatively small sample size. The potential limitations of using LLMs for generating requirements and domain labels are discussed, with a human evaluation provided to assess annotation quality. Future plans to address these limitations, including expanding to other programming languages and increasing dataset size, are outlined.

**Opportunities For Improvement:**

1. Expand the benchmark to include other programming languages beyond Python
2. Increase the sample size in future versions to enhance statistical robustness
3. Further validate the auto-generated annotations (requirements and domain labels)
4. Explore more advanced retrieval-augmented generation techniques
5. Investigate the reasons behind identified "comfort" and "strange" domains for different LLMs

**Relation To Prior Work:**

The authors clearly discuss how EvoCodeBench differs from previous contributions in code generation benchmarks. They provide a comprehensive comparison with existing benchmarks (Table 2) and explicitly address the limitations of prior work, particularly regarding data leakage and domain-specific evaluations. EvoCodeBench is effectively positioned as a novel contribution addressing important gaps in current code generation benchmarks.

**Summary And Contributions:**

This paper introduces EvoCodeBench, a new benchmark for evaluating Large Language Models (LLMs) in code generation. It addresses two main limitations of existing benchmarks: data leakage and lack of domain-specific evaluation. Key contributions:

1. An evolving benchmark updated every ~6 months to avoid data leakage
2. A domain taxonomy with 10 popular programming domains
3. Domain-specific evaluations, including a new Domain-Specific Improvement metric
4. EvoCodeBench-2403: First version with 275 samples from 25 recent repositories
5. Comprehensive evaluation of 8 popular LLMs, revealing insights into domain-specific performance

---

> ### Author Rebuttal · Authors · 2024-08-16
>
> ### **1. Response to C1 (small sample size...)**
>
> Thank you for the suggestion. Our response is in the **Response to C1** in the above **Response to All Reviewers**.
>
> ### **2. Response to C2 (Limited to Python...)**
>
> Thank you for the suggestion. Our response is in the **Response to C2** in the above **Response to All Reviewers**.
>
> ### **3. Response to C3 (More advanced RAG techniques...)**
>
> Thank you for the suggestion. **We have designed a retrieval-augmented baseline and evaluated it upon EvoCodeBench-2403.** This baseline considers the input requirement as a query and retrieves similar functions from the current repository. Because most programs in repositories are not equipped with documentation, we retrieve top-k (i.e., k = 5 in this paper) functions with similar names to the target function. More details are available in Appendix E.3. The results are shown in Table 9 in Appendix E.3. The performance of both LLMs is improved after introducing similar functions. We attribute improvements to relevant algorithms and dependencies in similar functions.
>
> Because this paper focuses on proposing an evolving benchmark, we leave more advanced retrieval-augmented generation techniques in future work.
>
> ### **4. Response to C4 (Reasons for comfort and strange domains...)**
>
> Thank you for the suggestion. Because most LLMs' pre-training data and pre-training settings are closed source, we cannot accurately determine the reason for comfort and strange domains. We inspect a potential reason is that LLMs' pre-training data mix is different. For example, GPT-4’s training data contains fewer repositories in the Internet domain, resulting in weak performance. In the future, we will use more advanced model interpretability techniques to explore the reasons for comfort and strange domains.

---

> ### Author Rebuttal · Authors · 2024-08-21
>
> Dear Reviewer ci62,
>
> Thank you again for reviewing our paper and for the valuable feedback. We have made every effort to address your concerns. As the rebuttal period is coming to an end, we are eager to know any additional comments or questions you may have. Thank you again for your time!
>
> Sincerely,
>
> Authors

---

### Author Rebuttal · Authors · 2024-08-16

## Responses to All Reviewers

We would like to thank all the reviewers for their comprehensive reviews and constructive feedback. We respond here to the same questions raised by the reviewers.

### **1. Response to C1 (Limited data size of EvoCodeBench-2403)**

**EvoCodeBench-2403 has the capability to distinguish superior and inferior LLMs.** (1) As shown in Table 4, EvoCodeBench-2403 reveals the performance gaps between LLMs in real-world repositories. For example, GPT-4 performed best among the studied LLMs, followed by GPT-3.5 and DeepSeek-Coder-33B. In Lines 216-221, we also analyzed 50 error cases of LLMs and summarized the reasons for errors. (2) As shown in Tables 5 and 6, we also evaluate the performance of LLMs in specific domains. We found that the performance of LLMs varies in different domains. For example, DeepSeek-Coder-6.7B is comparable to GPT-3.5 in the software development domain. These evaluation results are helpful for practitioners in picking up superior LLMs in specific domains.

**The data size of EvoCodeBench-2403 is greater than many popular benchmarks.** As shown in Table 2, a mainstream benchmark - HumanEval, only consists of 164 samples. The latest repo-level benchmark, CoderEval, only includes 230 samples.  In contrast, our EvoCodeBench-2403 contains more samples, including 275 samples from 25 repositories.

**EvoCodeBench is an evolving benchmark that will be continually expanded in the future.** Nowadays, EvoCodeBench-2403 only considers repositories created from 2023-10 to 2024-03. In the future, we will expand EvoCodeBench by collecting new samples from the latest repositories. For example, we plan to release the second version, EvoCodeBench-2409, in October 2024, which covers repositories created from 2024-04 to 2024-09. EvoCodeBench-2409 contains over 600 samples and covers 10 programming domains.

### **2. Response to C2 (Only containing samples in Python)**

**Python is one of the most popular programming languages in software development.** Since its creation in the early 1990s, Python has been used to build websites and software and analyze data by an estimated 8.2 million users [1]. Thus, we chose to construct EvoCodeBench starting from Python. Besides, we noticed that existing LLMs (e.g., CodeGen-Mono and Code Llama-Python) and code generation benchmarks (HumanEval, ClassEval, and DS-1000) focus on Python. This phenomenon also validates our motivation.

**Constructing multilingual versions of EvoCodeBench is laborious, and we will leave them in future work.** Expanding EvoCodeBench to other programming languages is not trivial and requires solving many technical problems, such as extracting calling graphs, installing running environments, and extracting unit tests.  Therefore, we first released the version in Python and plan to release the versions In Java and C++ by 2025. The detailed release plan is in the response to C4.

[1] https://flatironschool.com/blog/python-popularity-the-rise-of-a-global-programming-language/

### **3. Response to C3 (The anonymization of personal information in EvoCodeBench)**

Thank you for the suggestion. (1) We only consider repositories with permissive licenses. We are allowed to modify and re-distribute these repositories. (2) We have checked repositories in EvoCodeBench-2403 and anonymized all personal information. Specifically, we use regular expressions to extract suspicious strings (e.g.,  phone numbers, URLs, and emails) in repositories and manually check and anonymize them. We will also anonymize personal information in future versions of EvoCodeBench.

### **4. Response to C4 (Maintain Plan of EvoCodeBench)**

In the future, we will work with the Alibaba Team to maintain EvoCodeBench until we deem it unnecessary (e.g., the model's performance reaches human expectations or there is a better alternative). Our plans are shown in the following Table. In Oct. 2024, we will release the second version - EvoCodeBench-2409. It will contain over 600 samples in Python and cover 10 domains. Each domain contains at least 50 samples. In future versions, we will expand the data size and cover more languages (e.g., Java and C++).

| Date | Version | Data Size | Language | Note |
|:---:|:---:|:---:|:---:|:---:|
| 2024/10 | EvoCodeBench-2409 | 600+ | Python | It is the second version of EvoCodeBench. It contains over 600 Python samples and covers 10 programming domains, each with at least 50 samples. |
| 2025/03 | EvoCodeBench-2503 | 1500+ | Python, Java | It is the third version of EvoCodeBench. Besides Python, it further introduces samples in Java. |
| 2025/10 | EvoCodeBench-2509 | 2000+ | Python, Java, C++ | It is the fourth version of EvoCodeBench. We plan to include samples in C++. |
| TBD | TBD | TBD | TBD | Continually updating EvoCodeBench and covering new domains |

---

### Decision · Program_Chairs · 2024-09-26

**Decision:**

Accept (Poster)

**Comment:**

Thanks for your submission to NeurIPS 2024. Overall, the reviewers recognize that the paper presents a well-constructed and novel benchmark for evaluating the code generation abilities of LLMs w.r.t domain-specific evaluations. The benchmark is aligned with the characteristics of real-world repositories, which further enhance the usability of the EvoCodeBench.  The reviewers agree that this paper makes important contributions to the community and is valuable to be accepted.

Note from PC: This year, the track has been incredibly competitive, which meant that many good papers had to be rejected. After careful discussion with the SACs we have concluded that this paper unfortunately cannot be accepted this time. This is the final decision, which cannot be appealed. We encourage the authors to incorporate feedback from reviewers and additional results / discussion provided during the author response period in their next submission.